



# A full-mission data set of H$_2$O and HDO columns from SCIAMACHY 2.3 µm reflectance measurements

Andreas Schneider, Tobias Borsdorff, Joost aan de Brugh, Haili Hu, and Jochen Landgraf

SRON Netherlands Institute for Space Research, Utrecht, the Netherlands

**Correspondence:** Andreas Schneider (a.schneider@sron.nl)

**Abstract.** A new data set of vertical column densities of the water vapour isotopologues H$_2$O and HDO from the Scanning Imaging Absorption Spectrometer for Atmospheric Chartography (SCIAMACHY) instrument for the whole mission period from January 2003 to April 2012 is presented. The data are retrieved from reflectance measurements in the spectral range 2339 nm to 2383 nm with the Shortwave Infrared CO Retrieval (SICOR) algorithm, ignoring atmospheric light scattering in the measurement simulation. The retrievals are validated with ground-based Fourier transform infrared measurements obtained within the Multi-platform remote Sensing of Isotopologues for investigating the Cycle of Atmospheric water (MUSICA) project. A good agreement for low-altitude stations is found with an average bias of $-3.6 \cdot 10^{21}$ molec cm$^{-2}$ for H$_2$O and $-1.0 \cdot 10^{18}$ molec cm$^{-2}$ for HDO. The a posteriori computed $\delta$D shows an average bias of $-8 \permil$, even though polar stations have a larger negative bias. The latter is due to large sensor noise of SCIAMACHY in combination with low albedo and high solar zenith angles. To demonstrate the benefit of accounting for light scattering in the retrieval, the quality of the data product fitting effective cloud parameters simultaneously with trace gas columns is evaluated in a dedicated case study for measurements round high altitude stations. Due to a large altitude difference between the satellite ground pixel and the mountain station, clear sky scenes yield a large bias, resulting in a $\delta$D bias of 125 ‰. When selecting scenes with optically thick clouds within 1000 m above or below the station altitude, the bias in a posteriori $\delta$D is reduced from 125 ‰ to 44 ‰. The insights from the present study will also benefit the analysis of the data from the new Sentinel 5 Precursor mission.

## 1  Introduction

Atmospheric water vapour is a trace gas which is important for the energy budget of the atmosphere. For instance, it is the strongest natural greenhouse gas and transports energy through latent heat (e. g. Kiehl and Trenberth, 1997; Harries, 1997). However, the uncertainties regarding the effects of water vapour in the energy balance of the atmosphere and regarding the interaction between water vapour and the atmospheric circulation are still large (e. g. Stevens and Bony, 2013). Measurements are expected to contribute to a better understanding and quantification of these processes. Especially observations of isotopologues of water are interesting, because the ratio provides information about the "history" of the sampled air parcel due to a temperature-dependent isotopic fractionation during evaporation and condensation caused by different vapour pressure and diffusion constants of the isotopologues (e. g. Dansgaard, 1964). For instance, when water is evaporated from a not too small





reservoir, heavy isotopologues are depleted. Water vapour is also depleted by condensation during the formation of clouds and precipitation. In all these cases the amount of depletion depends on temperature.

The ratio of the isotopologues HDO and $H_2O$ is for column densities $c_{HDO}$ and $c_{H_2O}$ defined by $R_D = c_{HDO}/c_{H_2O}$. The depletion is usually described by the relative difference of an observed ratio $R_D$ to a standard ratio $R_{D,std}$,

$$\delta D = \frac{R_D - R_{D,std}}{R_{D,std}} \cdot 1000\%_o .\tag{1}$$

This is a column $\delta D$, which some authors refer to as $\overline{\delta D}$ to distinguish it from $\delta D$ profiles. The commonly used standard ratio is Vienna Standard Mean Ocean Water (VSMOW), $R_{D,std} = 3.1152 \cdot 10^{-4}$.

Measurements of atmospheric water vapour isotopologues are rare. Observations are made in situ with aircraft and balloon (e. g. Rinsland et al., 1984; Stowasser et al., 1999; Ehhalt et al., 2005; Dyroff et al., 2010), in situ on ground (e. g. Wen et al., 2010; Bastrikov et al., 2014), and by (ground or space based) remote sensing techniques. Ground-based remote sensing of the water vapour isotopologues is usually performed by Fourier transform infrared (FTIR) instruments which observe the direct sun light in the infrared, determining the vertically integrated columns above the site. Many ground stations are organised in networks, such as the Total Carbon Column Observing Network (TCCON, tccondata.org) and the Network for the Detection of Atmospheric Composition Change (NDACC, www.ndacc.org). Within the latter, the Multi-platform remote Sensing of Isotopologues for investigating the Cycle of Atmospheric water (MUSICA) project (Schneider et al., 2016) provides a validation against in situ measurements. Satellite-based measurements have the advantage of global coverage. $H_2O$ and HDO was first retrieved from satellite data by Zakharov et al. (2004) using thermal infrared measurements from the Interferometric Monitor for Greenhouse gases (IMG) sensor aboard the Advanced Earth Observing Satellite (ADEOS). Later, these isotopologues were also inferred from the Tropospheric Emission Spectrometer (TES) on the Earth Observing System (EOS) Aura satellite (Worden et al., 2006), the Michelson Interferometer for Passive Atmospheric Sounding (MIPAS) aboard European Space Agency (ESA)'s environmental satellite (ENVISAT) (Steinwagner et al., 2007), the Infrared Atmospheric Sounding Interferometer (IASI) abord the MetOP satellite (Herbin et al., 2009), and the Greenhouse Gases Observing Satellite (GOSAT) (Boesch et al., 2013). Frankenberg et al. (2009) used the short-wave infrared (SWIR) band of the Scanning Imaging Absorption Spectrometer for Atmospheric Chartography (SCIAMACHY) instrument (Bovensmann et al., 1999) on ENVISAT to obtain three years of simultaneous measurements of $H_2O$ and HDO. This data set was extended to the period 2003–2007 by Scheepmaker et al. (2015). In the present study a new $H_2O$ and HDO data set for the whole mission period of SCIAMACHY is presented.

Recently, the new Tropospheric Monitoring Instrument (TROPOMI) aboard the Sentinel 5 Precursor satellite (Veefkind et al., 2012) was launched, which includes the SWIR in the same spectral range and with the same spectral resolution as SCIAMACHY, but at much better signal to noise ratio (SNR) and spatial resolution. It is expected that retrievals from this new instrument will provide $H_2O$/HDO data of unprecedented quality. The present study is also a preparation for the new mission, apart from providing a data set for the whole mission period of SCIAMACHY.

The new data set is retrieved using the Shortwave Infrared CO Retrieval (SICOR) algorithm as discussed by Scheepmaker et al. (2016); Landgraf et al. (2016); Borsdorff et al. (2016, 2017). SICOR is designed for the operational processing of carbon monoxide total column retrieval from TROPOMI measurements. The algorithm has two performance options. First, assuming a



strict cloud filtering, the retrieval ignores atmospheric scattering in the SWIR spectral range, and the atmospheric total column of $CH_4$, $H_2O$, HDO, and CO is inferred from the measurement. This approach, called non-scattering retrieval in this paper, is proposed by Scheepmaker et al. (2016) to generate the $H_2O$ and HDO full mission data product. In this study, the retrieval is applied to SCIAMACHY observations, and the $H_2O$ and HDO data product is validated against ground based measurements

from MUSICA. Alternatively, one can choose for a loose cloud clearing. Here, the optical depth and height of a scattering layer is retrieved from the $CH_4$ absorptions of the measurement using appropriate a priori information from chemical transport models (CTMs). This approach, hereinafter referred to as scattering retrieval, is the processing baseline for CO retrievals from SCIAMACHY and TROPOMI as discussed by Landgraf et al. (2016); Borsdorff et al. (2017). In this paper, this approach is evaluated for the retrieval of water vapour isotopologues by comparing ground based observations at high-altitudes with

collocated SCIAMACHY retrievals using observations with clouds at a height similar to that of the ground site.

The remainder of this article is structured as follows: In Sect. 2 the retrieval set-ups of both the non-scattering and scattering retrieval are given. Section 3 presents the validation of the non-scattering data product, whereas Sect. 4 discusses the potential added value of the scattering retrieval of the water isotopologues. Finally, a summary is given and conclusions are drawn in Sect. 5.

## 2  Retrieval method

The SICOR algorithm and its application to CO and HDO/$H_2O$ retrievals is discussed in detailed by Scheepmaker et al. (2016); Landgraf et al. (2016); Borsdorff et al. (2016, 2017). In this section its main features relevant in the context of this study are summarised.

SICOR provides two options to perform the trace gas column retrieval from SWIR radiance measurements. First, assuming

clear sky observations, atmospheric scattering is ignored in the forward simulation of the measurement. Subsequently the algorithm infers the total column of $H_2O$, HDO, $CH_4$ and CO together with the Lambertian surface albedo from SCIAMACHY's channel 8 measurements between 2338.5 nm and 2382.5 nm. This spectral fit window is an extension of the one proposed by Scheepmaker et al. (2015) including more absorption lines of HDO, which is beneficial for retrievals using measurements at the end of SCIAMACHY's lifespan (which was not considered by Scheepmaker et al. (2015)) where more and more pixels

ceased to work. Spectral absorption line data are taken from Scheepmaker et al. (2013); Rothman et al. (2009); Predoi-Cross et al. (2006). The atmospheric absorption within this spectral range is presented in Fig. 1. The trace gas column retrieval utilises the profile scaling approach as described in detail by Borsdorff et al. (2014). The a priori profiles of water vapour are adapted from the European Centre for Medium-Range Weather Forecasts (ECMWF) reanalysis product. Since the ECMWF data product does not provide the individual isotopologue profiles, $H_2O$ and HDO profiles are obtained from the water vapour

profile by scaling it with the standard abundance of $H_2O$ and HDO, respectively. A priori profiles for methane and CO are taken from TM5 simulations. Obviously, the assumption of clear sky observations requires a cloud clearing of the data, which is performed a posteriori to the retrieval. As proposed by Scheepmaker et al. (2015), only retrievals are considered which data



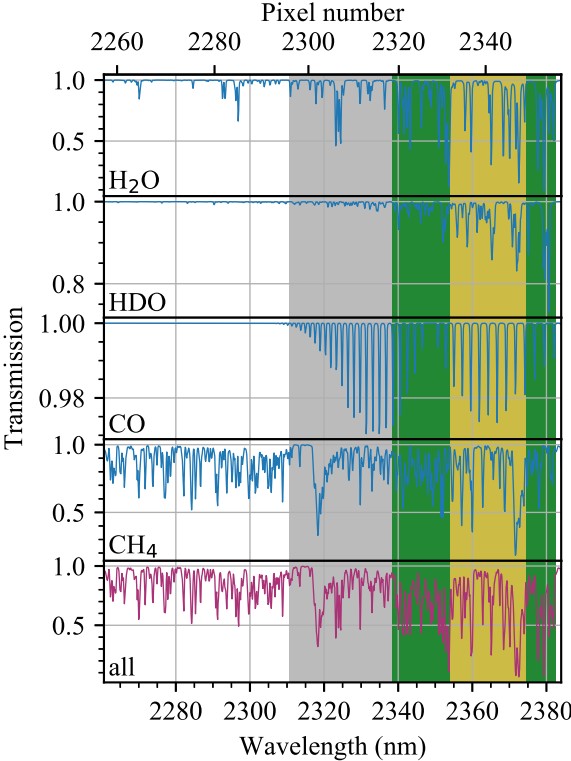

**Figure 1.** Simulation of atmospheric absorption in the spectral range of SCIAMACHY's channel 8 for the absorbers taken into account by the retrieval algorithm. The grey shading marks the retrieval window for CO used by Borsdorff et al. (2016, 2017), the yellow shading the window for $H_2O$/HDO used by Scheepmaker et al. (2015), the green shading the extension of that window used in this work.

product satisfy

$$0.9 < \frac{c_{CH_4}}{c_{CH_4,TM5}} < 1.1$$
$$0.7 < \frac{c_{H_2O}}{c_{H_2O,ECMWF}}$$

(2)

with the retrieved column $c_{CH_4}$ and $c_{H_2O}$ and the corresponding model predication $c_{CH_4,TM5}$ and $c_{H_2O,ECMWF}$, respectively. Moreover, due to radiometric performance issues of SCIAMACHY in the SWIR spectral range, the data has to be filtered with respect to outliers. Since water vapour in the troposphere is log-normally distributed (Schneider et al., 2006) and (as seen in our data) so is its uncertainty, the 15.9 and 84.1 percentiles $P_{15.9}$ and $P_{84.1}$ of the logarithmic $H_2O$ and HDO uncertainty data distributions $\log e$ are considered to define

$$\sigma = \frac{1}{2}(P_{84.1} - P_{15.9}) .$$

(3)



Only data within $5\sigma$ around the median $\mu$ of the logarithmic distribution are used. A similar filter is applied to the root mean square of the spectral fit residual, $c_{\mathrm{rms}}$, where data points within $6\sigma$ around the median $\mu$ are kept. Thus,

$$\mu_{e(\mathrm{H_2O})} - 5\sigma_{e(\mathrm{H_2O})} < \log e_{\mathrm{H_2O}} < \mu_{e(\mathrm{H_2O})} + 5\sigma_{e(\mathrm{H_2O})}$$

$$\mu_{e(\mathrm{HDO})} - 5\sigma_{e(\mathrm{HDO})} < \log e_{\mathrm{HDO}} < \mu_{e(\mathrm{HDO})} + 5\sigma_{e(\mathrm{HDO})} \tag{4}$$

$$\mu_{\mathrm{rms}} - 6\sigma_{\mathrm{rms}} < \log c_{\mathrm{rms}} < \mu_{\mathrm{rms}} + 6\sigma_{\mathrm{rms}}$$

Furthermore, the data are screened with respect to the number of iterations, $N_{\mathrm{iter}} \leq 12$, and the solar zenith angle (SZA), $\vartheta_{\mathrm{sz}} < 70°$.

Alternatively, SICOR allows to account for atmospheric scattering in the retrieval and so enhances the data yield of the retrieval. Based on the numerically very efficient two-stream radiative transfer model (Landgraf et al., 2016), the retrieval uses prior model information on CH$_4$ to infer the height $h_{\mathrm{cld}}$ and optical depth $\tau_{\mathrm{cld}}$ of a scattering layer together with the total column of CO, H$_2$O and HDO and the Lambertian surface albedo. Using this approach, our study considers for the first time the retrieval of the water vapour isotopologues for cloudy atmospheres and demonstrates the added value of the approach for the data product. The same quality filter as described above is used. The cloud filtering is detailed in Sect. 4.

The performance of SCIAMACHY suffered from an ice layer building up on the SWIR detector (e. g. Gloudemans et al., 2005). Among others, this caused additional stray light in the instrument, meaning in first approximation an additive radiometric bias to the measurement. To compensate this effect, a radiometric correction as described by Borsdorff et al. (2016) is applied. Hence, a radiometric offset with spectral dependence described by a third order polynomial is fitted for clear sky scenes above the Sahara region with a high albedo and so a high signal to noise ratio while fixing methane to the prior TM5

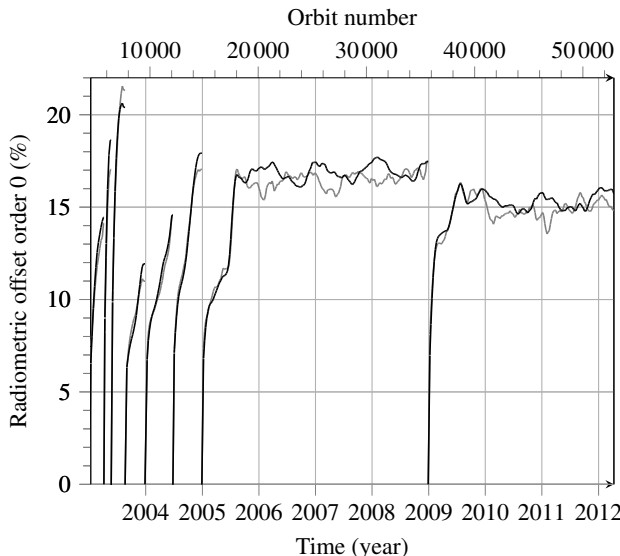

**Figure 2.** Zero-order radiometric offset retrieved from cloud-free SCIAMACHY spectra above the Sahara (black) and Australia (grey). The additive offset is given as percentage of the total radiance.





model information. Using the fitting window 2338.5–2382.5 nm, the time series of the spectrally constant radiometric offset is plotted in Fig. 2. It shows an increasing offset with growing ice layer which resets to zero at so-called decontamination events during which the detector was heated to remove the ice. Since 2005 no regular decontamination procedures were carried out with one additional decontamination at the beginning of 2009. Due to that the offset eventually stabilised. To demonstrate

that the calibration is globally applicable, the figure also shows the offset determined above Australia. The results are very similar and the deviation between both radiometric biases is less than 10 % when disregarding three outliers in the vicinity of decontamination events and a short period in the beginning of 2011 with a deviation of ∼14 %.

The strategy for the data product presented in this paper is to provide $H_2O$ and HDO columns for the whole period of the SCIAMACHY mission. The ratio $\delta D$ may be computed a posteriori from these data but is not a primary product.

## 3  Validation of the non-scattering retrieval of $H_2O$ and HDO

To validate the SCIAMACHY $H_2O$ and HDO total column, the observations are compared to ground-based FTIR measurements from 9 NDACC/MUSICA stations (Barthlott et al., 2017; Schneider et al., 2016, 2012), which are listed in Table 1. Here, high-altitude stations like Jungfraujoch (3580 m above sea-level (a. s. l.)) and Izaña (2367 m a. s. l.) are not considered because the satellite and ground based measurements probe different altitude ranges for these cases. For each site, all SCIAMACHY

observations within a radius of 800 km around the station passing the filter described in Sect. 2 are selected. Finally for comparison, both for MUSICA and SCIAMACHY data, monthly medians are taken, so that the data with different sampling can be compared. Months in which the number of individual measurements contributing to the median is less than 5 % of that in the month with most data are excluded to avoid biases by bad statistics. The loose spatio-temporal coregistration criteria are required to include a sufficient amount of measurements to overcome the large measurement noise of SCIAMACHY.

Figure 3 depicts a time series of monthly medians for the station Bremen. For all three quantities $H_2O$, HDO, and $\delta D$, the measurements of both instruments agree well. The bias, which is defined as the average of the difference between SCIA-

**Table 1.** List of MUSICA ground stations used for the validation.

| Station name | Latitude | Longitude | Altitude | Time period |
| --- | --- | --- | --- | --- |
| Eureka | 80.1 °N | 86.4 °W | 610 m | 2006–2014 |
| Ny Ålesund | 78.9 °N | 11.9 °E | 21 m | 2005–2014 |
| Kiruna | 67.8 °N | 20.4 °E | 419 m | 1996–2014 |
| Bremen | 53.1 °N | 8.9 °E | 27 m | 2004–2014 |
| Karlsruhe | 49.1 °N | 8.4 °E | 110 m | 2010–2014 |
| Jungfraujoch | 46.6 °N | 8.0 °E | 3580 m | 1996–2014 |
| Izaña | 28.3 °N | 16.5 °W | 2367 m | 1999–2014 |
| Wollongong | 34.5 °S | 150.9 °E | 30 m | 2007–2014 |
| Lauder | 45.1 °S | 169.7 °E | 370 m | 1997–2014 |

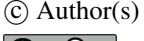



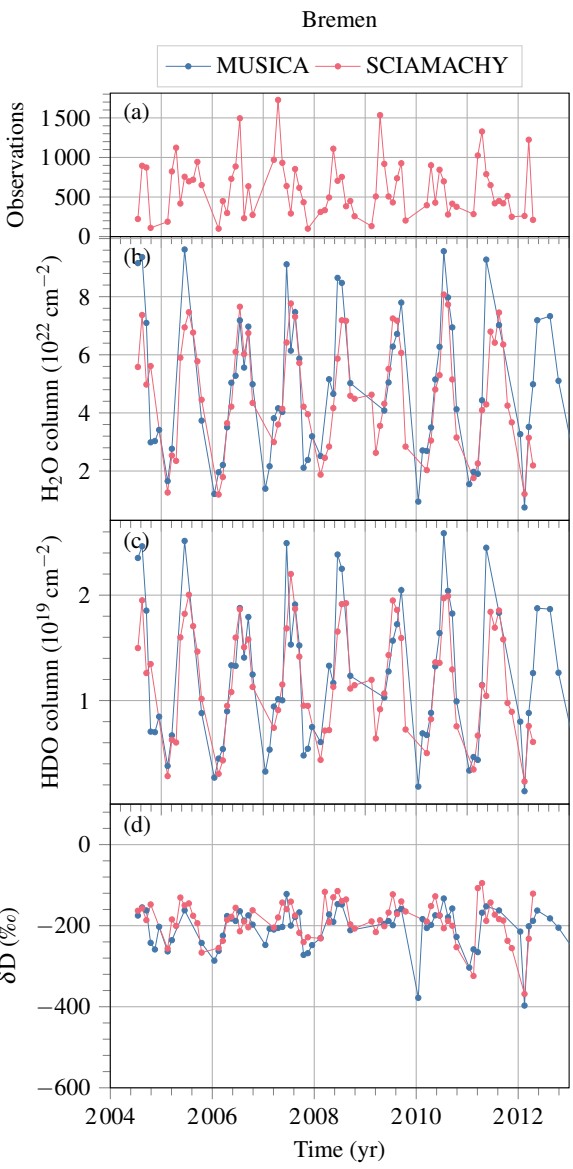

**Figure 3.** Time series of monthly medians of $H_2O$ **(b)**, HDO **(c)**, and $\delta D$ **(d)** for the MUSICA ground station Bremen (blue) and collocated SCIAMACHY observations for clear sky conditions (red). Panel **(a)** shows the number of SCIAMACHY measurements in each month.

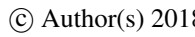



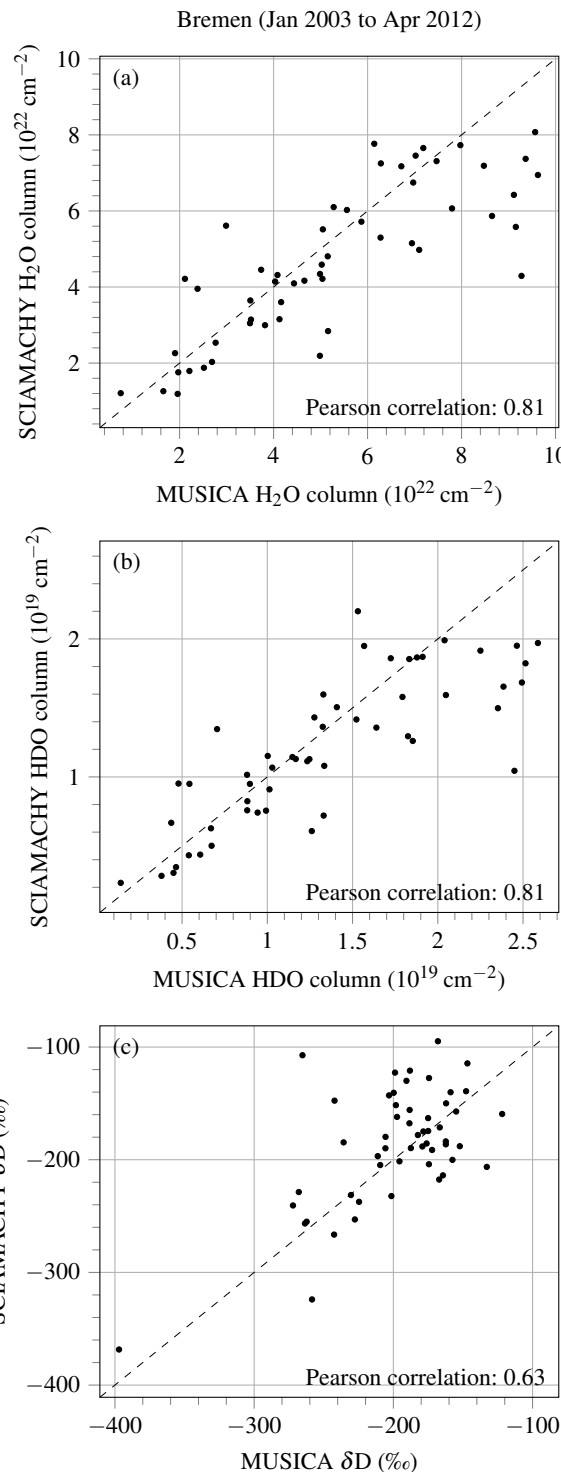

**Figure 4.** Correlation plots for monthly medians of MUSICA and SCIAMACHY observations of $H_2O$ **(a)**, HDO **(b)**, and $\delta D$ **(c)** at the station Bremen between January 2003 and April 2012.



MACHY and MUSICA results, is $-5.6 \cdot 10^{20}$ molec cm$^{-2}$, $-1.4 \cdot 10^{17}$ molec cm$^{-2}$, and $+11‰$, respectively. The time series also shows that the data set is homogeneous throughout the whole mission period. Figure 4 shows a clear correlation between MUSICA and SCIAMACHY measurements for all three quantities with a Pearson correlation coefficient is 0.81, 0.81, and 0.63 for H$_2$O, HDO, and $\delta$D.

The statistics for all low-altitude MUSICA stations are presented in Fig. 5. For H$_2$O and HDO, the correlation is good (between 0.74 and 0.97) with small biases for all stations (on average $-3.6 \cdot 10^{21}$ molec cm$^{-2}$ and $-1.0 \cdot 10^{18}$ molec cm$^{-2}$, respectively). Also for $\delta$D a good correlations between SCIAMACHY and MUSICA data is achieved for most stations, but it is only mediocre for Kiruna, Wollongong, and Lauder. The reason is that the seasonality in $\delta$D is weak for those stations. To demonstrate that the low correlation coefficient does not indicate bad results, the time series for Lauder (for which the correla-

tion is lowest) is shown in Fig. 6. The polar stations Eureka and Ny Ålesund have a relatively high bias in $\delta$D. Although the number of observations is large due to orbits converging at polar latitudes, the results are dominated by difficult measurement conditions with low surface albedo and high solar zenith angles in combination with SCIAMACHY's high sensor noise, which result in imprecise results. The average bias in $\delta$D over all stations is $-8‰$.

     The bias is relatively stable over the mission time. When computing the bias for both halves of the mission from 2003 to

2007 and from 2008 to 2012, the difference is relatively small for most stations. On average the relative difference in bias in H$_2$O between both halves is 20 %, with a larger difference for Bremen where it changes from $-2.9$ to $-8.2 \cdot 10^{21}$ molec cm$^{-2}$. On the other hand, the bias in $\delta$D does not change in Bremen. The mean change in the bias of $\delta$D is 11 %. Only for Ny Ålesund the bias in $\delta$D becomes significantly worse from $-24‰$ for 2003–2007 to $-52‰$ for 2008–2012. The latter is attributed to the degradation of the instrument which especially plays a role for difficult measurement geometries.

To demonstrate the coverage of the non-scattering retrieval data set, world plots of H$_2$O, HDO, and $\delta$D averaged over the first and second half of the mission period are presented in the left and right panels of Fig. 7, respectively. In H$_2$O and HDO the latitudinal gradient with more water vapour abundance in the tropics and dryer air in polar regions is clearly visible. Above mountain ranges such as the Andes and the Himalayas, the water vapour abundance is reduced. This effect is discussed in more detail in the next section in connection with high-altitude stations. A continental effect with dryer air inland can also be seen,

e. g. in North America, North Africa and Asia. In $\delta$D, the features described by Scheepmaker et al. (2015) and Frankenberg et al. (2009) are mostly reproduced. On the large scale, $\delta$D decreases from the equator to the poles, which has already been mentioned by Dansgaard (1964). High altitudes also show a strong depletion in $\delta$D. The continental effect is visible, for example, in North and South America, Asia, and even Australia. Enhanced $\delta$D above the Red Sea seen by Frankenberg et al. (2009) is also reproduced. More importantly, the comparison between the left and right panels shows that the data set is

consistent throughout the mission time.



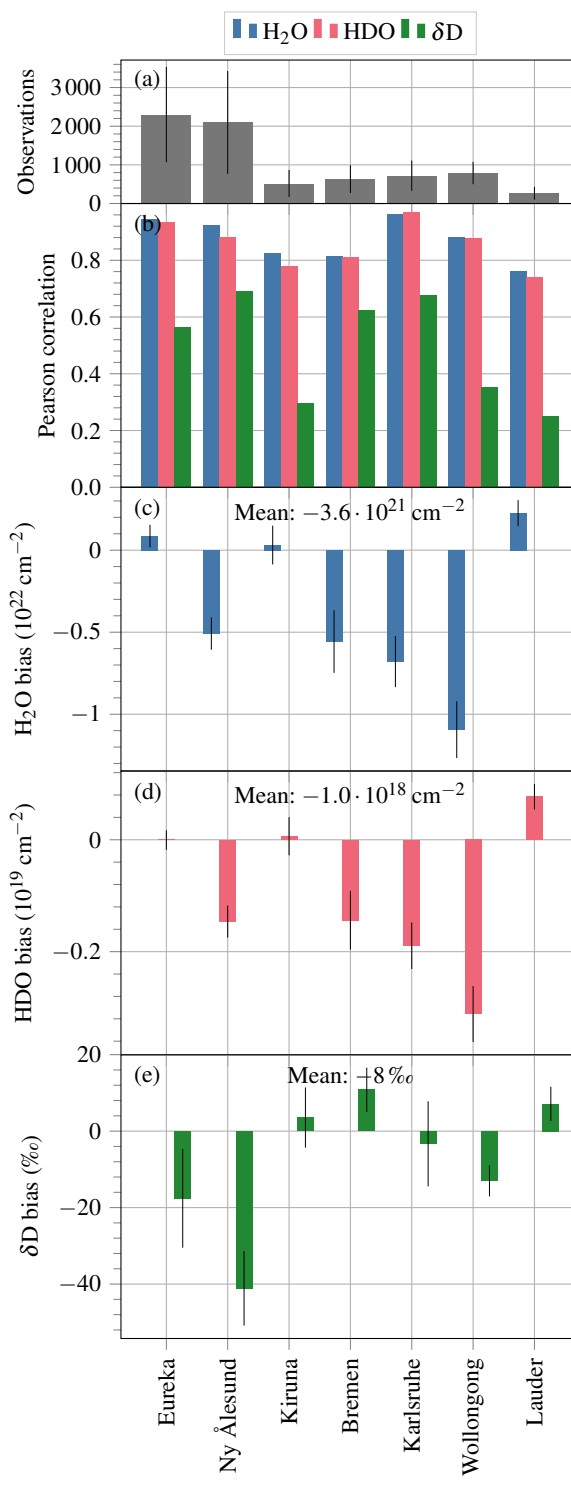

**Figure 5.** Statistics for the validation of the non-scattering retrievals: **(a)** average number of measurements per month and its standard deviation. **(b)** Pearson correlation coefficient between MUSICA and SCIAMACHY monthly averages of $H_2O$ (blue), HDO (red), and $\delta D$ (green). **(c)** bias of $H_2O$ and its standard error. **(d)** bias of HDO and its standard error. **(e)** bias of $\delta D$ and its standard error.





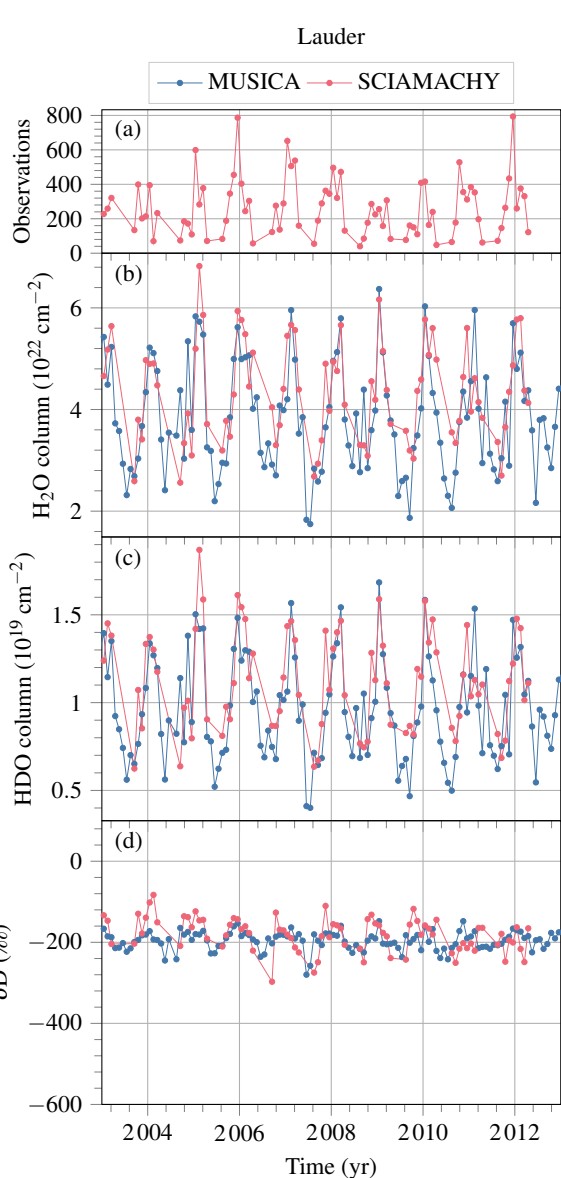

**Figure 6.** Same as Fig. 3, but for the station Lauder





**Figure 7.** $H_2O$ column **(a, b)**, HDO column **(c, d)**, and $\delta D$ **(e, f)** on a $1° \times 1°$ grid averaged over the first half from 2003 to 2007 (left) and the second half from 2008 to 2012 (right) of the SCIAMACHY mission



## 4 The scattering retrieval of H$_2$O and HDO

To demonstrate the benefit of accounting light scattering by clouds in the retrieval of H$_2$O and HDO, the corresponding water vapour isotopologue product of SCIAMACHY is compared with MUSICA measurements at elevated sites above 2000 m a. s. l. Here, significant differences between the mean surface height of the SCIAMACHY ground pixels and the surface elevation of the ground site hampers the verification of the measurements using clear sky observations. However, when retrieving cloud height and scattering optical depth jointly with the water columns, satellite and ground based measurements of similar altitude sensitivity can be selected. This is demonstrated for the MUSICA site at Jungfraujoch (3580 m a. s. l.) by selecting collocated SCIAMACHY observations for clear sky and cloudy sky conditions using the selection criteria in Table 2.

A time series of clear sky observations in the left panel of Fig. 8 shows a large bias in the SCIAMACHY measurements relative to MUSICA of $4.3 \cdot 10^{22}$ molec cm$^{-2}$ in H$_2$O, $1.1 \cdot 10^{19}$ molec cm$^{-2}$ in HDO, and 140‰ in $\delta$D. Atop the mountain at Jungfraujoch, only the partial column above 3580 m is observed, while SCIAMACHY in a collocation radius of 800 km around the site observes the whole column above the ground pixel, which elevation is much lower than Jungfraujoch for most scenes. With the large vertical gradient of water vapour abundance in the lower troposphere, this leads to the observed bias. Moreover, the column-averaged $\delta$D has lower values over mountains due to the HDO depletion with height.

To correct the SCIAMACHY observations for this altitude difference, the collocated ECMWF water vapour profile is scaled to the total column observed by SCIAMACHY, and subsequently the scaled profile is truncated at the height of Jungfraujoch. The vertical integration of this truncated profile provides the altitude corrected H$_2$O and HDO columns depicted in the centre panels of Fig. 8. Obviously, the correction eliminates the large biases for H$_2$O and HDO for the most part; the biases are reduced by more than a factor of 400 to $1.0 \cdot 10^{20}$ molec cm$^{-2}$ and by more than a factor of 36 to $3.0 \cdot 10^{17}$ molec cm$^{-2}$, respectively. However, for $\delta$D computed from the corrected columns the bias remains due to the assumption about the same relative vertical distribution of both isotopologues.

Next, cloudy scenes with cloud height around the station altitude are evaluated, since for these the result for the whole column is dominated by the part above the cloud due to the measurement sensitivity. The sensitivity of the retrieved column

**Table 2.** Selection criteria for clear sky and cloudy sky conditions using scattering retrieval. Here $h_\mathrm{s}$ is the altitude of the MUSICA ground site and max(SNR) is the maximum of the signal-to-noise ratio (SNR) of a spectrum.

| Quantity | filter |
|---|---|
| **Clear sky filter** | |
| Cloud height | $h_\mathrm{cld} < 1000$ m |
| Cloud optical thickness | $\tau_\mathrm{cld} < 0.2$ |
| SNR | $5 \leq \max(\mathrm{SNR}) \leq 150$ |
| **Filter for optically thick clouds** | |
| Cloud height | $h_\mathrm{s} - 1000\,\mathrm{m} \leq h_\mathrm{cld} \leq h_\mathrm{s} + 1000\,\mathrm{m}$ |
| Cloud optical thickness | $\tau_\mathrm{cld} > 3$ |





**Figure 8.** Time series of monthly medians similar as shown in Fig. 3, but for cloud retrievals near the high-altitude station Jungfraujoch (3580 m a. s. l.). The left panels **(a, d, g, j)** show clear sky measurements without altitude correction, the centre panels **(b, e, h, k)** the same measurements with altitude correction, the right panels **(c, f, i, l)** observations with optically thick clouds within an altitude range 1000 m above and below the station height. Please note that in the left panels the $H_2O$ and HDO axes are different than in the centre and right panels, as indicated by the axis ticks.





to changes in the vertical distribution of the water vapour isotopologues is described by the column averaging kernel (e. g. Rodgers, 2004). An example for clear sky and cloudy measurements is shown in Fig. 9. A detailed discussion of the column averaging kernel is given by Borsdorff et al. (2014). In case of an optically thick cloud (in the example at 3.5 km altitude), the retrieval is sensitive above the cloud but insensitive below the cloud, which underlines the selection criteria for cloudy sky

observations in Table 2. Furthermore, to compare MUSICA data with cloudy SCIAMACHY observations, it is important to realise that cloudy conditions usually involve a higher humidity than clear sky conditions. The ground based measurements are performed at the latter and so MUSICA data are first interpolated to the time of the satellite observations using ECMWF, similar as described by Borsdorff et al. (2016, Sect. 4.1) (compare Fig. 8e and f). A height difference between SCIAMACHY ground pixel and MUSICA station is corrected as described above. The right panel of Fig. 8 depicts the result. The bias in

$\delta$D between SCIAMACHY and MUSICA is significantly reduced and the difference between panels k and l demonstrates the altitude dependence of $\delta$D. Only satellite measurements with similar altitude sensitivity as the ground based observation yield good agreement.

Figure 10 presents the statistics for the cloud retrievals above both high-altitude stations Jungfraujoch and Izaña. The correlation in $H_2O$ and HDO is generally good for all evaluations, however it is small for $\delta$D, although it increases somewhat

for the cloud retrieval. This is due to the noise in $\delta$D being nearly as large as its seasonality. When looking at the bias, the behaviour already seen in the time series is confirmed. It is large for all three quantities for uncorrected clear sky observations. The altitude correction reduces significantly the bias in $H_2O$ and HDO, while no change occurs to $\delta$D. The latter is significantly reduced by regarding scenes with clouds at an altitude near the station height. The bias in $H_2O$ and HDO increases slightly when going from clear sky to cloudy scenes, but is still much lower than for the uncorrected case. This may be explained by

the differences in cloud height compared to the station height accepted by the data selection.

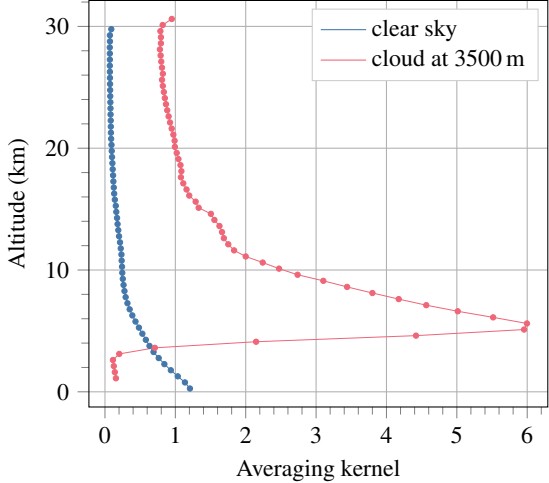

**Figure 9.** Example of averaging kernels for a clear sky measurement (blue) and a scene with optically thick clouds (red)





**Figure 10.** Like Fig, 5, but for cloud retrievals for high-altitude stations. The left column shows results for clear sky measurements without altitude correction, the centre column those for clear sky observations with altitude correction, and the right column those for altitude-corrected observations with optically thick clouds within the station height plus/minus 1000 m.





For cloud retrievals from SCIAMACHY observations at polar latitudes the data quality is reduced. Due to the limited radiometric performance of SCIAMACHY, the retrieval of cloud properties from methane absorption feature is hampered by the low SNR for these challenging observation geometries and the dominance of water vapour absorption in the considered spectral range.

Due to these instrumental problems at polar latitudes, the non-scattering retrieval is preferred for the global SCIAMACHY data set. For the new TROPOMI instrument, however, the cloud retrieval is expected to work for all latitudes, because it has a much better signal to noise ratio and a much better radiometric calibration, and the size of the ground pixel (and thus the averaging over areas with potentially varying cloud cover) is much smaller. The last point is also important since water vapour is known to vary considerably both spatially and temporally. In this light, the current study is also a preparation for the future

investigations with TROPOMI.

## 5    Summary and conclusions

In this paper an new data set of satellite observations of column densities of the water vapour isotopologues $H_2O$ and HDO from the SCIAMACHY instrument is presented, which spans the whole mission period from January 2003 to April 2012. This is an addition by more than 4 years compared to Scheepmaker et al. (2015). To this end, the degradation of the instrument has to be

mitigated by enlarging the retrieval window compared to Scheepmaker et al. (2015) and by eliminating the radiometric offset via a calibration with the Sahara region as natural target. Atmospheric light scattering is ignored in the forward simulation of the measurements. Additionally, the possibility to infer simultaneously effective cloud parameter and the water vapour abundance is considered in a dedicated case study for measurements around high-altitude stations.

      Due to the high sensor noise of SCIAMACHY, an adequate averaging in time and/or space has to be performed to obtain

reasonable results. For instance, monthly medians are meaningful when averaging over a circle with 800 km radius. More spatial resolution such as $1° \times 1°$ global coverage can be achieved when sacrificing temporal resolution by averaging over several years.

      The satellite observations are validated against low-altitude ground-based FTIR measurements from the MUSICA project within the NDACC network. In general, the agreement is good. The Pearson correlation coefficient for $H_2O$ and HDO is

between 0.74 and 0.97, and the average bias is only $-3.6 \cdot 10^{21}$ molec cm$^{-2}$ for $H_2O$ and $-1.0 \cdot 10^{18}$ molec cm$^{-2}$ for HDO. For $\delta$D, the correlation is also good for most stations (between 0.56 and 0.79) with three exceptions where the seasonal variation is small. The bias is low except at polar stations (Eureka and Ny Ålesund). The decreased performance at high latitudes stems from imprecise measurements caused by the large sensor noise of SCIAMACHY in combination with difficult measurement geometries. The average bias in $\delta$D over all stations is $-8‰$, but when excluding Eureka and Ny Ålesund it is only $+1‰$.

High-altitude stations (Jungfraujoch and Izaña) are treated separately in a case study with a retrieval that takes scattering into account and additionally infers cloud parameters. Since both instruments observe different altitude ranges, the bias is large for clear sky measurements with $3.8 \cdot 10^{22}$ molec cm$^{-2}$ for $H_2O$, $1.0 \cdot 10^{19}$ molec cm$^{-2}$ for HDO and $125‰$ for $\delta$D.





To correct for the altitude effect, partial columns above the station height are computed for SCIAMACHY by cutting an ECMWF water vapour profile scaled to the retrieved column. This yields good agreement between SCIAMACHY and MUSICA in $H_2O$ and HDO, the mean bias is reduced to $-2.2 \cdot 10^{20}\,\mathrm{molec\,cm^{-2}}$ and $3.4 \cdot 10^{17}\,\mathrm{molec\,cm^{-2}}$, respectively. The bias for the a posteriori $\delta D$ cannot be corrected that way, however, because the profiles for $H_2O$ and HDO have the same shape, neglecting the increasing depletion with height.

The average bias in $\delta D$ is drastically reduced to $44\,‰$ by considering scenes with optically thick clouds between $1000\,\mathrm{m}$ below and $1000\,\mathrm{m}$ above the station height and applying the altitude correction, since for such scenes the retrieval is sensitive predominantly above the cloud. For this approach, the SCIAMACHY results in all three quantities $H_2O$, HDO, and $\delta D$ agree well with MUSICA. The corresponding biases in $H_2O$ and HDO are $1.4 \cdot 10^{21}\,\mathrm{molec\,cm^{-2}}$ and $4.5 \cdot 10^{17}\,\mathrm{molec\,cm^{-2}}$, respectively.

The scattering retrievals work well for low and mid-latitudes, as demonstrated in the case study for high-altitude stations, but are unreliable for polar latitudes. This is caused by the limited radiometric performance of SCIAMACHY in combination with difficult observation geometries. For the newly launched TROPOMI instrument, which measures in the SWIR range with similar resolution than SCIAMACHY but with a much better signal to noise ratio and a much higher spatial resolution, the method is expected to work for all latitudes. Moreover, the inferred columns are expected to have a higher precision, eliminating the necessity to average over long times or large areas. Thus future investigations will concentrate on TROPOMI.

*Data availability.* The full-mission SCIAMACHY $H_2O$/HDO data set from the non-scattering retrieval described in this paper is available for download at ftp://ftp.sron.nl/pub/pub/DataProducts/SCIAMACHY_HDO/. The MUSICA data was downloaded from ftp://ftp.cpc.ncep.noaa.gov/ndacc/MUSICA/.

*Competing interests.* The authors declare that they have no conflict of interest.

*Acknowledgements.* SCIAMACHY is a joint project of the German Space Agency DLR and the Dutch Space Agency NSO with contributions from the Belgian Space Agency. The data of the ground-based measurements have been provided by the project MUSICA, which has been funded by the European Research Council under the European Community's Seventh Framework Programme (FP7/2007–2013)/ERC grant agreement number 256961.



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
