# Peer review of "A full-mission data set of $\text{H}_2\text{O}$ and $\text{HDO}$ columns from SCIAMACHY 2.3 $\mu\text{m}$ reflectance measurements"

_Atmospheric Measurement Techniques, 2017_

## Referee Comment (RC1) · Anonymous Referee #2 · 18 Mar 2018

This paper evaluated the vertical column densities of H2O and HDO retrieved from SCIAMACHY for the whole mission (2003 – 2012) using the non-scattering retrieval mode of the Shortwave Infrared CO Retrieval (SICOR) algorithm. The SCIAMACHY results were compared against the ground-based FTIR measurements from MUSICA. Reasonably high correlation coefficients and relatively small biases for H2O and HDO were obtained at the stations. δD was calculated a posteriori from H2O and HDO, and generally had larger errors. A comparison between the first and second half of the mission showed that the datasets were self-consistent.

This paper also assessed the benefit of the scattering retrieval mode of SICOR using a case study where the SCIAMACHY results for cloudy scenes were compared with high-altitude ground-based data. The most significant improvement was a clear reduction of the error in δD.

In general, the paper clearly conveyed the scientific findings. The results can provide guidance for SCIAMACHY data users. The method used can be useful for current and future missions.

The following suggestions may be considered:
(1) Use symbols in Figure 7 to show the locations of the MUSICA stations.
(2) Over the ocean, how do SCIAMACHY H2O data compare against satellite microwave data or other references?
(3) Provide a conversion factor from molec cm^-2 to mm for H2O, since the latter is commonly used in atmospheric science.
(4) Page 18 Line 14, change "similar resolution than SCIAMACHY" to something like "a similar spectral resolution to that of SCIAMACHY".

---

## Referee Comment (RC2) · Anonymous Referee #1 · 30 Mar 2018

This study presents the retrievals of columnar water vapor isotopes (H2O and HDO) from the SCIAMACHY. Using an algorithm that was developed to retrieve H2O/HDO along with CO, authors implement additional spectral range to the algorithm and apply it to the satellite's entire mission period. Ground-based measurements at several sites are used for validating retrieved H2O/HDO columnar concentration and depletion. The study also explores the potential benefit by jointly retrieving cloud height and scattering optical depth. They demonstrate that the inclusion of cloud scattering can correct biases in H2O/HDO retrievals at an elevated site. This study addresses an important topic. It fall into the scope of AMT journal. Retrieval approach and validation efforts are well designed and well presented. The results are sufficiently discussed. I have only a

few comments, but none of them I believe is major.

Specific comments: P6, L15: Here an area with radius of 800 km surrounded the site is used to sample the satellite data, which is a substantially large area. While it is mentioned in the paper to "include a sufficient amount of measurements", it may need some additional justifications on how such a big area can well represent the site.

P9, L19: Can the author explain more on how the degradation "especially plays a role for difficult measurement geometries"? And what does the mean by the "difficult measurement geometries"?

P15, L14: "increases somewhat" -> "increases slightly"

―――――――――――――――――――

---

## Author Comment (AC1) · 4 May 2018

**Author response to review RC1 by Anonymous Referee #2**

*This paper evaluated the vertical column densities of H2O and HDO retrieved from SCIAMACHY for the whole mission (2003–2012) using the non-scattering retrieval mode of the Shortwave Infrared CO Retrieval (SICOR) algorithm. The SCIAMACHY results were compared against the ground-based FTIR measurements from MUSICA. Reasonably high correlation coefficients and relatively small biases for H2O and HDO were obtained at the stations. δD was calculated a posteriori from H2O and HDO, and*

[Figure]

*generally had larger errors. A comparison between the first and second half of the
mission showed that the datasets were self-consistent.*

*This paper also assessed the benefit of the scattering retrieval mode of SICOR using a
case study where the SCIAMACHY results for cloudy scenes were compared with high-
altitude ground-based data. The most significant improvement was a clear reduction
of the error in $\delta D$.*

*In general, the paper clearly conveyed the scientific findings. The results can provide
guidance for SCIAMACHY data users. The method used can be useful for current and
future missions.*

We thank the reviewer for the positive feedback. In the following, the individual points
are quoted in italics part by part, and our response is given below. Page and line
numbers in the response refer to the revised version of the manuscript.

Individual comments

(1) *Use symbols in Figure 7 to show the locations of the MUSICA stations.*

   We have changed the figure and the corresponding caption accordingly (p. 12).

(2) *Over the ocean, how do SCIAMACHY H2O data compare against satellite mi-
   crowave data or other references?*

   We agree that a validation over the ocean would be beneficial. However, the avail-
   ability of data to compare against is very limited. To our knowledge, H2O/HDO
   data from satellite observations using the microwave spectral range do not ex-
   ist; water isotopologue retrievals are only available in the shortwave infrared and
   thermal infrared spectral ranges. An intercomparison with other satellite data
   would involve caring for different sensitivities and footprints, e. g. by using av-
   eraging kernels etc. This is outside the scope of this paper. Thus, we rely on

well-established ground data. The observations over Tenerife Island at Izaña may be representative of oceanic conditions.

(3) *Provide a conversion factor from molec cm⁻² to mm for H2O, since the latter is commonly used in atmospheric science.*

In our manuscript we have added the following sentence stating conversion factors (p. 7 ll. 3–4):

The conversion factor to precipitable water in mm is $2.99 \cdot 10^{-22}$ mm molec$^{-1}$ cm$^2$ for H2O and $3.16 \cdot 10^{-22}$ mm molec$^{-1}$ cm$^2$ for HDO.

(4) *Page 18 Line 14, change "similar resolution than SCIAMACHY" to something like "a similar spectral resolution to that of SCIAMACHY".*

The sentence is changed as follows (p. 18 l. 13 ff.):

For the newly launched TROPOMI instrument, which measures in the SWIR range with a spectral resolution similar to that of SCIAMACHY but with a much better signal to noise ratio and a much higher spatial resolution, . . .

---

## Author Comment (AC2) · 4 May 2018

**Author response to review RC2 by Anonymous Referee #1**

*This study presents the retrievals of columnar water vapor isotopes (H2O and HDO) from the SCIAMACHY. Using an algorithm that was developed to retrieve H2O/HDO along with CO, authors implement additional spectral range to the algorithm and apply it to the satellite's entire mission period. Ground-based measurements at several sites are used for validating retrieved H2O/HDO columnar concentration and depletion. The study also explores the potential benefit by jointly retrieving cloud height and scatter-*

*ing optical depth. They demonstrate that the inclusion of cloud scattering can correct biases in H2O/HDO retrievals at an elevated site. This study addresses an important topic. It fall into the scope of AMT journal. Retrieval approach and validation efforts are well designed and well presented. The results are sufficiently discussed. I have only a few comments, but none of them I believe is major.*

We thank the reviewer for the positive review. The specific comments are responded to below. Page and line numbers in the response refer to the revised version of the manuscript.

Specific comments

*P6, L15: Here an area with radius of 800 km surrounded the site is used to sample the satellite data, which is a substantially large area. While it is mentioned in the paper to "include a sufficient amount of measurements", it may need some additional justifications on how such a big area can well represent the site.*

Considering a validation of individual measurements, the reviewer is right that the comparison between a MUSICA measurement and a SCIAMACHY observation within an 800 km radius around the ground site may be dominated by representation errors. A single MUSICA measurement cannot estimate the SCIAMACHY observation due to the high variability of atmospheric water vapour. However, the low precision of individual SCIAMACHY observations additionally requires a temporal averaging of both SCIAMACHY and MUSICA data. Therefore, according to our results, representation errors appear to cancel out to a large extend. Thus we have added the following sentence to our manuscript (p. 6 ll. 19–21):

Even though water vapour may change substantially over small distances, the results suggest that representation errors due to the large spatial collocation area average out in the monthly medians and their statistics.

*P9, L19: Can the author explain more on how the degradation "especially plays a role for difficult measurement geometries"? And what does the mean by the "difficult measurement geometries"?*

Difficult measurement conditions are high solar zenith angles and low surface albedos, as stated in the previous paragraph in the manuscript (p. 9 ll. 11–12 in the discussed version). We have added a reference to that and a sentence elaborating on why the degradation especially plays a role for these (p. 9 ll. 16–19 in the updated manuscript):

The latter is attributed to the degradation of the instrument which especially plays a role for difficult measurement conditions as described above. Low sun and low albedo result in low signal to noise ratio and thus higher error sensitivity. Outside polar latitudes easier measurement conditions can make up for the degradation.

*P15, L14: "increases somewhat" -> "increases slightly"*

Changed (p.15 l. 14).

––––––––––––––––––––––––––––––––